# Comparison of hip function and quality of life of total hip arthroplasty and resurfacing arthroplasty in the treatment of young patients with arthritis of the hip joint at 5 years

Matthew L Costa,[1] Juul Achten,[1] Pedro Foguet,[2] Nicholas R Parsons,[3] On behalf of the Young Adult Hip Arthroplasty team

[1]Oxford Trauma, NDORMS, Kadoorie Centre, University of Oxford, John Radcliffe Hospital, Oxford, UK
[2]University Hospitals Coventry and Warwickshire NHS Trust, Coventry, UK
[3]Warwick Medical School, University of Warwick, Coventry, UK

**Correspondence to**
Professor Matthew L Costa;
matthew.costa@ndorms.ox.ac.uk

## ABSTRACT

**Objective** To compare the medium-term clinical effectiveness of total hip arthroplasty and resurfacing arthroplasty.

**Design** Single centre, two-arm, parallel group, assessor blinded, randomised controlled trial with 1:1 treatment allocation.

**Setting** A large teaching hospital in England.

**Participants** 122 patients older than 18 years with severe arthritis of the hip joint, suitable for resurfacing arthroplasty of the hip. Patients were excluded if they were considered to be unable to adhere to trial procedures or complete questionnaires.

**Interventions** Total hip arthroplasty (replacement of entire femoral head and neck); hip resurfacing arthroplasty (replacement of the articular surface of femoral head only, femoral neck remains intact). Both procedures replaced the articular surface of the acetabulum.

**Outcomes** The outcome measures were hip function assessed using the Oxford Hip Score (OHS) and health-related quality of life assessed using the EuroQol (EQ-5D). Patients were followed up annually for a minimum of 5 years. Outcome data were modelled using the generalised estimating equation methodology to explore temporal variations during follow-up.

**Results** 60 patients were randomly assigned to hip resurfacing arthroplasty and 62 to total hip arthroplasty. 95 (78%) of the 122 original study participants provided data at 5 years. There was a small decrease in both hip functions and quality of life in both groups of patients each year during the 5-year follow-up period. However, there was no evidence of a significant difference between treatments group in the OHS (P=0.333) or the EQ-5D (P=0.501).

**Conclusions** We previously reported no difference in outcome in the first year after surgery. The current medium-term results also show no evidence of a difference in hip function or health-related quality of life in the 5 years following a total hip arthroplasty versus resurfacing arthroplasty.

**Trial registration number** ISRCTN33354155. UKCRN 4093.

### Strength and limitations of this study

► The main strength of this trial is that it is pragmatic, with a large number of surgeons using a variety of different hip arthroplasty implants and their own preferred surgical technique.
► Other strengths include the use of validated patient-reported outcome tools, a relatively large number of participants for this type of trial and the high levels of complete follow-up data.
► The key limitation of this trial was that the patients themselves could not be blind to their type of hip arthroplasty.

## INTRODUCTION

For older patients with severe arthritis of the hip, several designs of total hip arthroplasty (THR) have shown excellent long-term results in terms of both function and value for money.[1] However, in younger and more active patients, there is an approximate 50% failure rate at 25 years for traditional implants.[2] Modern THR designs may improve on these results,[3] but the search for new, more durable forms of arthroplasty continues. One option is resurfacing arthroplasty of the hip (RSA).[4]

Resurfacing implants are more expensive than traditional (metal and plastic) THR designs and there are potential complications associated with RSA compared with THR—most importantly the risk of fracture of the neck of the femur.[5] However, the first clinical results showed that in selected patients, 98% of RSA implants were still functioning at 5 years[6]; which is as good as any of the existing THR designs.[1] Furthermore, by preserving the patient's own proximal femoral anatomy, it was suggested that RSA may provide more physiological hip movement. Early clinical outcomes indicated that RSA provides

improved hip function when compared with THR.[7 8] Other studies[7 9] reported that patients having RSA had higher activity levels after the procedure and were more likely to be involved in activities such as running and heavy manual labour. However, these studies were not randomised clinical trials.

In 2012, we reported the 1 year results of a randomised controlled trial to compare the clinical effectiveness THR and RSA in patients with severe arthritis of the hip.[10] There was no evidence of a difference in functional outcomes or health-related quality of life at 1 year. Data regarding cost-effectiveness were reported separately.[11] In this report, we provide the minimum 5-year follow-up data from the same cohort of patients randomised into the original trial.

## PATIENTS AND METHODS

This was a single-centre, two-arm, parallel group, assessor-blind randomised controlled trial with 1:1 treatment allocation conducted in the UK. Full details of the protocol have been described previously.[12] A summary of the methodology follows below.

### Study population

In this pragmatic trial, participants were eligible if they were over 18 years of age, medically fit for an operation and suitable for a RSA—patients suitable for RSA are also suitable for THR. Patients were only excluded from the study if there was evidence that the patient would be unable to adhere to trial procedures or complete questionnaires. To maintain independence between observed outcomes, if a recruited patient required a contralateral hip arthroplasty during the trial period the second hip was not included in the study.

### Recruitment and randomisation of participants

The Warwick Arthroplasty Trial (WAT) opened in May 2007 and 126 patients were recruited between August 2007 and February 2010 from hip arthroplasty clinics in a single UK hip arthroplasty centre. Eligible patients gave written informed consent. They were randomised on a 1:1 basis to receive either a THR or an RSA. Treatment allocation was determined using a computer-generated, randomised number sequence and stratified by the supervising orthopaedic surgeon to balance any potential surgeon effects. After patients consented to participate in the trial, an independently administered, secure randomisation service was alerted by telephone of a new enrolment. The randomisation officer provided the surgeon's secretary with the patient's treatment allocation, thereby keeping the research associates, who consented patients and collected outcome data, blinded to the allocated treatment.

### Interventions

Each patient had the allocated surgery according to the preferred technique and implants of the operating surgeon. Other perioperative interventions, such as prophylactic antibiotics and thromboprophylaxis, were the same for all patients. After the operation, all patients underwent the same standardised rehabilitation plan, including range-of-movement exercises followed by muscle strengthening exercises. Unless the operating surgeon specifically advised otherwise, all patients were fully weight bearing immediately.

In a THR, the femoral head was removed along with most of the femoral neck. The femoral shaft was exposed to open up the femoral canal. The femoral component was then inserted into the canal and the articulating femoral head was placed onto the neck of the femoral component. The choice of components and bearing surfaces was left to the discretion of the operating surgeon, as per their usual clinical practice.

In a resurfacing arthroplasty, the articular surfaces of the femoral head was removed but the neck was left in situ. The femoral component (cap) was then impacted onto the patient's own femoral neck. All resurfacing arthroplasties of the hip used metal-on-metal bearing surfaces, but the choice of surgical approach, implant size and positioning was left to the discretion of the operating surgeon.

In both forms of arthroplasty, the acetabulum is prepared and the acetabular component inserted into the socket.

### Outcome measurements

The primary outcome measure for this medium-term follow-up study was hip function, assessed using the OHS,[13] and the secondary outcome was the EuroQol 5D (EQ-5D) health-related quality of life utility score (HRQoL).[14] Each outcome was collected annually by self-reported postal questionnaire. All complications related to the hip arthroplasty were recorded during the course of the trial.

On completion of the main phase of the trial, the trial steering group recommended to remove the additional outcomes collected during main phase (Harris Hip Score, Disability Rating Index, physical activity level and resource use) to reduce the burden on the participants and to optimise retention rates.

### Statistical analysis

Longitudinal data (ie, the time course of measurements at yearly intervals) were modelled using generalised estimating equations (GEEs) to explore the population-averaged effects of time and operative treatment on function and HRQoL.[15] A first-order autoregressive working correlation model was adopted to account for within-subject temporal correlations of untransformed OHS and EQ-5D outcomes, which were assumed to be approximately normally distributed and related to the linear predictor using the identity link function. Statistical significance was assessed at the 5% level and CIs of estimated temporal trends in OHS and EQ-5D were constructed by non-parametric bootstrapping. Models

**Table 1** Baseline characteristics of 122 participants by received intervention; data shown are means (SD)[10]

| Treatment | RSA (n=60) | THR (n=62) |
|---|---|---|
| Sex | F=24 and M=36 | F=27 and M=35 |
| Age (years) | 56.5 (6.9) | 56.7 (7.0) |
| BMI (kg/m$^2$) | 28.4 (6.2) | 28.9 (4.8) |
| Oxford Hip Score | 19.0 (7.7) | 19.3 (7.9) |
| EQ-5D score | 0.31 (0.35) | 0.36 (0.36) |
| EQ-5D VAS | 56.1 (23.4) | 57.6 (24.2) |

BMI, body mass index: EQ-5D, EuroQoL Five Dimensions; RSA, resurfacing arthroplasty of the hip; THR, total hip arthroplasty; VAS, visual analogue scale.

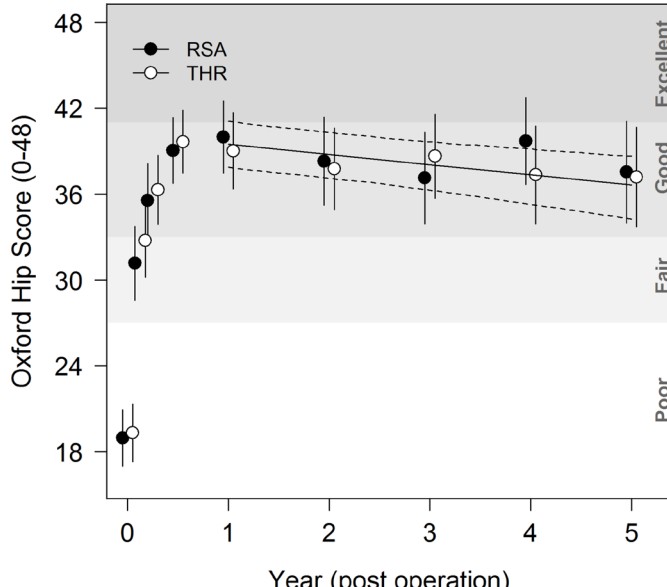

**Figure 1** Temporal trends in Oxford Hip Score for participants from operation to year 5. RSA, resurfacing arthroplasty of the hip; THR, total hip arthroplasty.

with smaller a quasilikelihood information criteria (QIC) were considered to be better descriptors of the data.[15] Differences in complication rates between groups were assessed using Fisher's exact test. All analyses were undertaken in R V.3.2.2.[16]

## RESULTS

The WAT study recruited 126 participants, 95% of whom had a primary diagnosis with osteoarthritis. Of the 126 participants recruited into the WAT study, 4 never had an operation, leaving 122 in total available for follow-up during the original trial and in the extended (medium-term) follow-up study that we report here. The baseline characteristics of these 122 participants are shown in table 1.

The following medium-term follow-up results are based on the treatment received (per-protocol analysis), in contrast to the previously reported study results which were based on the allocated intervention (ie, intention-to-treat analysis). The amount of missing data at each time-point is shown in table 2. A small number of patients died during follow-up, did not respond to attempts to contact them or actively asked to be withdrawn from the study. In total, 95 (78%) of the 122 original study participants provided data at 5 years. We have no reason to believe that withdrawals or loss was related to the intervention,

so we will assume for purposes of analysis that data were missing completely at random.

## OHS

OHS scores at each year postoperation (from year 1 to 5) were strongly positively autocorrelated; that is, a high OHS at year 1 was predictive of a high OHS in subsequent years. The OHS scores decreased over time from year 1 to year 5 (GEE z-test of regression coefficients; P=0.003, with the regression coefficient −0.70). However, this decline was unlikely to be clinically relevant in the medium term, occurring at only approximately 0.70 OHS units per year. Figure 1 shows the temporal trends in OHS for participants from operation to year 5.

Adding model terms to account for treatment groups (RSA vs THR) did not lead to a reduction in QIC, and model parameters showed that there was no evidence of a significant difference between treatment groups (P=0.333) during follow-up, and the rate of decline of OHS

**Table 2** Patient follow-up at yearly intervals postoperation

| | Year of follow-up | | | | | | | | | |
|---|---|---|---|---|---|---|---|---|---|---|
| | 1 | | 2 | | 3 | | 4 | | 5 | |
| Patient status | RSA | THR | RSA | THR | RSA | THR | RSA | THR | RSA | THR |
| Died | 0 | 0 | 0 | 0 | 1 | 0 | 1 | 0 | 1 | 0 |
| Missing* | 0 | 0 | 7 | 5 | 6 | 2 | 7 | 5 | 11 | 11 |
| Responded† | 60 | 62 | 53 | 57 | 53 | 59 | 50 | 56 | 45 | 50 |
| Withdrawn | 0 | 0 | 0 | 0 | 0 | 1 | 2 | 1 | 3 | 1 |
| Total | 60 | 62 | 60 | 62 | 60 | 62 | 60 | 62 | 60 | 62 |

*Missing data, lost to follow-up.
†Patient followed-up.
RSA, resurfacing arthroplasty of the hip; THR, total hip arthroplasty.

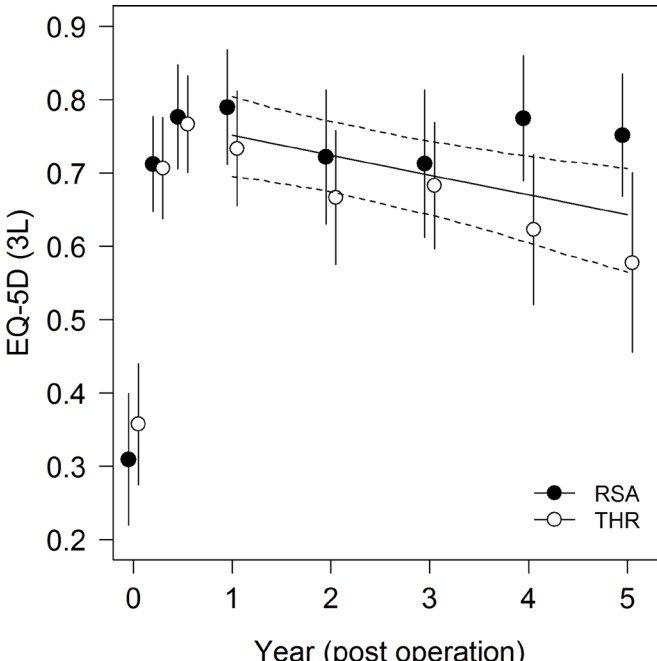

**Figure 2** Temporal trends in EQ-5D for participants from operation to year 5. EQ-5D, EuroQoL Five Dimensions; RSA, resurfacing arthroplasty of the hip; THR, total hip arthroplasty.

score did not differ between treatment groups (P=0.317). Full results of analyses are shown in online supplementary appendix A1.

### HRQoL

There was also evidence for a significant temporal change in EQ-5D (GEE z-test of regression coefficients; P=0.002); EQ-5D scores decreased over time from year 1 to year 5 (regression coefficient –0.027). However, again the decline in EQ-5D scores was relatively small at approximately 0.027 units per year.

EQ-5D scores showed considerable variability during follow-up (figure 2). Adding model terms to account for treatment groups did not lead to a reduction in QIC, and model parameters showed that there was no evidence for a statistically significant difference between treatments groups (P=0.501). The rate of decline of EQ-5D scores did not differ between treatment groups (P=0.236). Full results of analyses are shown in online supplementary appendix A2.

### Complications

During the 5-year follow-up, one patient in the RSA group had a revision arthroplasty and three in the THR group

**Table 3** Revisions at 5 years by treatment group

|  | RSA | THR | Total |
|---|---|---|---|
| Revised | 1 | 3 | 4 |
| Unrevised | 44 | 47 | 91 |
| Total | 45 | 50 | 95 |

RSA, resurfacing arthroplasty of the hip; THR, total hip arthroplasty.

(table 3). Two other patients in the THR group suffered a dislocation of the hip but did not require revision surgery. One patient in the RSA group had an aspiration of the hip joint but did not require revision surgery.

Fisher's exact test provides no evidence of a statistically significant difference in revision rates between treatment groups (P=0.619).

### DISCUSSION

Our previously reported randomised clinical trial found no evidence of a difference in hip function between patients having THR versus RSA for severe arthritis of the hip joint during the first year following surgery.[10] This medium-term follow-up study continues to show no difference in hip function at 5 years. Similarly, there was no difference in HRQoL. The number of further complications after the first year was low in both groups, but one patient in the RSA group and three in the THR group required revision arthroplasty surgery.

Only a few randomised trials have been performed comparing THR with the resurfacing technique. The first studies focused on the technical aspects of the procedure, such as the position of the implants or the amount of bone removed during the resurfacing procedures.[17 18] Three trials investigated clinical outcomes for resurfacing arthroplasty compared with a specific type of THR, namely metal-on-metal THR.[19–21] All of these trials showed little difference in functional outcome between the groups in the first 1 to 2 years after surgery. Each of the trials included plans to perform longer term follow-up of the participants. However, subsequent, widely reported concerns regarding the adverse effects of metal debris from metal-on-metal bearing surfaces have made it difficult to interpret any later results from these trials[22 23]; particularly because the functional deficits associated with adverse reactions to metal debris seemed to be greater in one group (THR) than the other (RSA).[24]

A further randomised trial[25] looked at early muscle strength in 43 patients and found greater muscle strength deficits in the RSA group. In contrast, a trial of 80 patients with dysplastic acetabula found improved early range of movement in the RSA group although with no difference in functional hip scores.[26]

The only longer term follow-up study of resurfacing versus conventional bearing-surface THR showed no difference between groups with regard to OHS or quality life.[27] More patients in the resurfacing group were involved in impact activities. However, this study contained only 24 randomised patients.

The main strength of this trial is that it is entirely pragmatic, with a relatively large number of surgeons using a variety of different hip arthroplasty implants and their own preferred surgical technique. Although the patients were recruited from only one centre, the large number of surgeons involved and the variety of implants is likely to reflect practice in the wider surgical community. Other strengths include the use of

validated patient-reported outcome tools, a relatively large number of participants and the high levels of complete follow-up data.

The key limitation of this trial was that the patients themselves were not blind to their type of hip arthroplasty. Patients undergoing RSA in the UK have generally been given a different preoperative information sheet and surgical consent from than those having a THR; this reflects the existing evidence regarding the different risk/benefit profile of the two procedures.

How do the results of this trial inform the debate about resurfacing arthroplasty of the hip? This trial failed to show any evidence that resurfacing arthroplasty provides improved hip function or greater quality of life when compared with THR over 5 years. Given the new requirements for surveillance of metal-on-metal hip arthroplasties,[28] the higher rate of revision surgery for RSA recorded on the UK national joint registry[29] and increased costs associated with RSA,[30] it seems increasingly difficult to justify the use of this technology. We will, however, continue to review the patients in this trial with a further report planned at a minimum of 10 years.

**Acknowledgements** We would like to thank Becky Kearney, Katie McGuinness, Helen Richmond, Phil Jones, Kate Dennison, Zoe Buckingham, Troy Douglin and Catherine Richmond for their invaluable assistance in recruitment and data collection for this trial. We would like to thank Chris McCarthy, Chris Bridle, Ceri Jones, Tim Friede and Steve Krikler for their clinical, trials and regulatory expertise in the TSC and DMC for this trial. Finally, we would like to thank all the patients for the time and effort they gave to this trial.

**Collaborators** This trial was conducted on behalf of the Young Adult Hip Arthroplasty team at University Hospitals Coventry and Warwickshire NHS Trust: Matt Costa, Pedro Foguet, Udai Prakash, Damian Griffin, Richard King, Steve Krikler and Gavin Pereira.

**Contributors** MLC, JA and NRP designed the study and analysed and interpreted the trial data. PF managed the recruitment and follow-up of the patients. MLC, JA and NRP planned and wrote the first draft of the paper which was subsequently revised by all authors. All authors read and approved the final manuscript. MLC acts as guarantor of the paper.

**Funding** This work was funded by the National of Institute Health Research, Research for Patient Benefit under grant number PB-PG-0706-10080 and was supported bythe National Institute for Health Research (NIHR) Oxford Biomedical Research Centre . This trial was co-sponsored by the University of Warwick and University Hospitals Coventry and Warwickshire National Health Service Trust. This manuscript presents independent research commissioned by the NIHR.

**Disclaimer** The views expressed are those of the authors and not necessarily those of the NHS, the NIHR or the Department of Health.

**Competing interests** None declared.

**Patient consent** Not required.

**Ethics approval** This study has been reviewed by the Coventry Research Ethics Committee under reference number 07/Q2802/26 - approval was awarded on the 9th of May 2007. Further approval was obtained from the research and development department of UHCW NHS trust. The research carried out is in compliance with the Helsinki Declaration, and all patient provided informed consent prior to be included into the study.

**Provenance and peer review** Not commissioned; externally peer reviewed.

**Data sharing statement** No additional unpublished data are available.

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
