## [Reviewer comments · BMJ Open]

ARTICLE DETAILS

TITLE (PROVISIONAL)	Comparison of hip function and quality of life of Total Hip Arthroplasty and Resurfacing Arthroplasty in the Treatment of Young Patients with Arthritis of the Hip Joint at 5 years.
AUTHORS	Costa, Matthew; Achten, Juul; Foguet, Pedro; Parsons, Nicholas

VERSION 1 – REVIEW

REVIEWER	Akihiro Sudo Department of Orthopaedic Surgery, Mie University Graduate School of Medicine, Japan
REVIEW RETURNED	25-Sep-2017

GENERAL COMMENTS	The authors reported that the medium-term clinical results show no evidence of a difference in hip function or quality of life in the five years after total hip replacement versus resurfacing arthroplasty. However, there are several things to be cleared for acceptance of the publication. 1. The authors use “total hip arthroplasty” and “total hip replacement”. Please correct.2. After having shown an abbreviation, the authors use the full spelling. Please correct.3. In Discussion, the authors explain the abbreviation again. Please correct.4. Table 1: The data of 126 participants are not important. Please switch to the data of 122 participants.5. Figure 2 is not cited in the text.6. Please clarify the reasons of revision.7. The authors should describe conclusion at the end of Discussion.8. Reference 10, 11, 12 are wrong. Please correct.
---

REVIEWER	Szilard Nemes University of Gothenburg Sweden
REVIEW RETURNED	27-Sep-2017

GENERAL COMMENTS	The manuscript by Costa and collaborators compares the clinical effectiveness of total hip replacement and resurfacing arthroplasty. The manuscript is well written and reads well. I have some observations and questions that I hope the authors will consider addressing. Introductions The first paragraph of the introduction suggest to the readers that THA might not be beneficial to younger patients as the long term failure rate is high. Reading between the lines ones gets the feeling that the alternative, resurfacing surgery might be better. However the reference for that information is from 1999, so it is based on more than 40 year old data. It's rather obsolete, those findings likely does not apply for modern implants. Methods Perhaps this is due to my unfamiliarity with RCT studies but I'm struggling to fully grasp how the randomisation was done. Reading the Study Protocol did help, but I would prefer to see a more vivid description. Where the patient randomized to RSA/THA or surgeons whom prefer one or the other type of surgery? Results The participants in both arms are rather young. The proportion 60 years or younger is around 20 % in UK. The inclusion criteria does not considers age. Is there any reason for that? The authors deemed that a yearly decrease in of 0.7 for the OHS and 0.027 for the EQ-5D index is not clinically relevant. I tend to disagree, a 7 point decrease or of the OHS or 0.27 for the EQ-5D index in 10 years is a rather big loss. Despite lack of statistical significance, there is a rather worrying trend. Patients with THA show a rapid decline if HRQoL. The 5 year value drop below 0.6. This is in stark contrast with a recent observational study from Sweden (Bengtsson 2017 Acta Orthopaedica 88(5): 484-489). Swedish THA whom were in average 10 years older show little loss of HRQoL after 6 years and much higher values Having a EQ-5D index value below 0,6 five years after operation for an around 60 years old person gives reason for concern.
---

REVIEWER	Matthias Knobe RWTH Aachen University, Germany
REVIEW RETURNED	29-Sep-2017

GENERAL COMMENTS	This is a prospective randomised study looking at outcome parameter (score of function and quality of life) of middle-aged patients after hip arthroplasty. There was a randomisation to receive either a THR (total hip arthroplasty) or a RSA (resurfacing arthroplasty). Patients were followed up annually for a minimum of five years. As a result, there was no difference in function and in health-related quality of life in the five years. In general, the manuscript is well written with clear methods, a large number of participants and high levels of complete follow-up data. The study confirmed the results of their previous study in the first year after surgery. I have only minor suggestions and some questions: 1. For what reason there was a change of implant allocation in some cases? You have to mention this in the limitation section (as a kind
---

	of selection bias). 2. How many surgeons were in each study arm? Hip resurfacing can be a challenge sometimes - were there some specialists for this operation? 3. You reports 3 hip dislocations in the THR group. Have you an explanation for this complication? Without a revision (why?), had these patients a poorer outcome than the rest? 4. You stated a full weight-bearing standard after the operation, unless the surgeon advised otherwise. How many cases were affected by this? Were there differences in the study arms? 5. In the result section you mentioned a decrease of the EQ-5D scores from 1 to 5 years (Line 21-22). However, in figure 2 this decline is evident only for THR patients. Can you specify the statements in more detail?
--	--

REVIEWER	Lauren Barnett Arthritis Research UK Primary Care Centre, Keele University, UK.
REVIEW RETURNED	30-Oct-2017

GENERAL COMMENTS	1. I'm happy to see GEE's used properly, so well done! 2. I'd like to see more detail about the model you fitted, more specifically a coefficient table. Whilst p-values and comparison statistics are more understandable to a medical audience, it would be reassuring to see some model results. 3. Another minor note is justification of these methods. Why have you used GEEs and not multilevel models? Why did you use Fisher's exact test and not an alternative? 4. I'd also add what software you used at the end of the analysis section. 5. Adding more detail about the model you used, for instance the link function. 6. In the methods section of your abstract you should state that you used GEEs. 7. One more minor point, and one I won't insist on, is adding a small sensitivity analysis. You state that "within-subject temporal correlations...were assumed to be approximately normally distributed." Would the results change if you assumed they weren't? Adding a sensitivity analysis would show the robustness of your model to changes in underlying assumptions.
---

VERSION 1 – AUTHOR RESPONSE

Reviewer: 1

The authors reported that the medium-term clinical results show no evidence of a difference in hip function or quality of life in the five years after total hip replacement versus resurfacing arthroplasty. However, there are several things to be cleared for acceptance of the publication.

1. The authors use "total hip arthroplasty" and "total hip replacement". Please correct.

Done

2. After having shown an abbreviation, the authors use the full spelling.

Please correct.

Response: Done

3. In Discussion, the authors explain the abbreviation again.

Please correct.

Corrected

4. Table 1: The data of 126 participants are not important. Please switch to the data of 122 participants.

Response: Table 1 has been replaced with details of the n = 122 participants in the medium-term follow-up study.

5. Figure 2 is not cited in the text.

Response: Figure 2 is now properly referred to in text.

6. Please clarify the reasons of revision.

Response: I'm afraid that we do not have the details of the revision procedures, only that the components were revised c.f. further surgery involving retention of the original THR implants

7. The authors should describe conclusion at the end of Discussion.

Response: We are not quite sure what the reviewer means re this comment. We believe that we have drawn conclusions based upon the data presented, but are happy to take further advice if required.

8. Reference 10, 11, 12 are wrong. Please correct.

Response: Apologies, these have been corrected

Reviewer: 2

The manuscript by Costa and collaborators compares the clinical effectiveness of total hip replacement and resurfacing arthroplasty.

The manuscript is well written and reads well.

I have some observations and questions that I hope the authors will consider addressing.

Introductions

The first paragraph of the introduction suggest to the readers that THA might not be beneficial to younger patients as the long term failure rate is high. Reading between the lines ones gets the feeling that the alternative, resurfacing surgery might be better. However the reference for that information is from 1999, so it is based on more than 40 year old data. It's rather obsolete, those findings likely does not apply for modern implants.

Response: This reference is indeed from 1999, but we consider this a seminal paper re the long-term performance of THR which still holds true today – the benchmark against which more modern implants are judged.

Methods

Perhaps this is due to my unfamiliarity with RCT studies but I'm struggling to fully grasp how the randomisation was done. Reading the Study Protocol did help, but I would prefer to see a more vivid description. Where the patient randomized to RSA/THA or surgeons whom prefer one or the other type of surgery?

Response: We have clarified this

Results

The participants in both arms are rather young. The proportion 60 years or younger is around 20 % in UK. The inclusion criteria does not consider age. Is there any reason for that?

Response: The trial was designed to reflect current practice in the UK, where Resurfacing was generally offered to younger patients. However, we did not include an age limit in the eligibility criteria to allow for older but still very high-demand patients who may have been considered for Resurfacing.

Comment: The authors deemed that a yearly decrease in of 0.7 for the OHS and 0.027 for the EQ-5D index is not clinically relevant. I tend to disagree, a 7 point decrease or of the OHS or 0.27 for the EQ-5D index in 10 years is a rather big loss.

Comment: Despite lack of statistical significance, there is a rather worrying trend. Patients with THA show a rapid decline in HRQoL. The 5 year value drop below 0.6.

This is in stark contrast with a recent observational study from Sweden (Bengtsson 2017 *Acta Orthopaedica* 88(5): 484-489). Swedish THA whom were in average 10 years older show little loss of HRQoL after 6 years and much higher values. Having a EQ-5D index value below 0,6 five years after operation for an around 60 years old person gives reason for concern.

I'm afraid that we cannot really comment on the differences in HQoL described in other studies, although we agree with the reviewer that these are interesting.

Response: Regarding the fall in HQoL noted during our study; the fall within each year was relatively small but we agree with the reviewer that this accumulates to a clinically important fall over the course of the five year follow-up period and have amended the manuscript accordingly.

Reviewer: 3

This is a prospective randomised study looking at outcome parameter (score of function and quality of life) of middle-aged patients after hip arthroplasty. There was a randomisation to receive either a THR (total hip arthroplasty) or a RSA (resurfacing arthroplasty). Patients were followed up annually for a minimum of five years. As a result, there was no difference in function and in health-related quality of life in the five years.

In general, the manuscript is well written with clear methods, a large number of participants and high levels of complete follow-up data. The study confirmed the results of their previous study in the first year after surgery.

I have only minor suggestions and some questions:

1. For what reason there was a change of implant allocation in some cases? You have to mention this in the limitation section (as a kind of selection bias).

Response: All data in this manuscript is presented by treatment received, not allocated. WAT main study manuscript provides more details of the original trial (Costa ML, Achten J, Parsons N, Edlin RP, Foguet P, Prakash U, Griffin DR. Total hip arthroplasty versus resurfacing arthroplasty in the treatment of patients with arthritis of the hip joint: a single centre, parallel group, assessor blinded, randomised controlled trial. *BMJ* 2012; 344:e2147).

2. How many surgeons were in each study arm? Hip resurfacing can be a challenge sometimes. Were there some specialists for this operation?

Response: Again, full details have previously been provided in the previous publication in *BMJ*.

3. You reports 3 hip dislocations in the THR group. Have you an explanation for this complication? Without a revision (why?), had these patients a poorer outcome than the rest?

Response: Two patients in the THR group had a dislocation but did not undergo revision surgery. I'm afraid that we do not have details regarding why revision surgery was not offered although first-time dislocations of the hip are not always treated with revision surgery in the UK, provided they are not associated with component loosening or radiological evidence of component wear.

4. You stated a full weight-bearing standard after the operation, unless the surgeon advised otherwise. How many cases were affected by this? Were there differences in the study arms?

Response: Details of the rehabilitation are provided in the previous paper in the BMJ. We do not have the data to show how many patients had restrictions placed upon their weight-bearing, but it is standard practice in the UK to allow patients to fully weight-bear after any hip replacement so the number of patients affected is likely to have been very small.

5. In the result section you mentioned a decrease of the EQ-5D scores from 1 to 5 years (Line 21-22). However, in figure 2 this decline is evident only for THR patients. Can you specify the statements in more detail?

Response: The EQ-5D scores were highly variable, so it is difficult to draw any strong inferences from the available data. The statistical modelling suggests that the treatment differences do not yet approach significance. As we accumulate more data in subsequent years we will be able to make a more definitive statement on whether this is a 'real' and important treatment effect or simply natural (and normal) variation amongst participants.

Reviewer: 4

1. I'm happy to see GEE's used properly, so well done!
2. I'd like to see more detail about the model you fitted, more specifically a coefficient table.

Response: Whilst p-values and comparison statistics are more understandable to a medical audience, it would be reassuring to see some model results.

Response: Full results (output) from model fitting have been added in a new appendix.

3. Another minor note is justification of these methods. Why have you used GEEs and not multilevel models? Why did you use Fisher's exact test and not an alternative?

Response: The choice of appropriate model is driven in large part by the scientific question being addressed. In a clinical trial we are often more interested in differences for the subject (participant) and for instance in predicting individual (subject-specific) trajectories. Marginal models such as generalized estimating equations (GEE) are appropriate when the main focus of an analysis and resultant inference is on population-averaged effects, which is what we are primarily interested in here. How do the outcomes for the study population evolve over time? GEEs model within-subject (temporal) associations using an empirically determined covariance matrix and are particularly easy to implement for simple settings such as ours where there is a single source of clustering (i.e. follow-up time). Clearly, mixed-effects (multilevel) models may be more appropriate if there are complex (possibly nested) factors that result in short-term effects on outcomes; e.g. a recruiting centre or

operating surgeon in an RCT. In longer term follow-up studies, such as ours, these issues are likely to be negligible and no longer of much interest, so fitting marginal (population-averaged) models seems much more appropriate. As overall counts of complications was small (<5), Fisher's exact test was preferred to a chi-squared test

4. I'd also add what software you used at the end of the analysis section.

Response: Details added.

5. Adding more detail about the model you used, for instance the link function.

Response: Details added. The canonical identity link function was used.

6. In the methods section of your abstract you should state that you used GEEs.

Response: Details added.

7. One more minor point, and one I won't insist on, is adding a small sensitivity analysis. You state that "within-subject temporal correlations...were assumed to be approximately normally distributed." Would the results change if you assumed they weren't? Adding a sensitivity analysis would show the robustness of your model to changes in underlying assumptions.

OHS and EQ-5D scores are almost universally assumed to be approximately normally distributed for purposes of analysis. We agree that that it would be reassuring to undertake some sensitivity analyses. However, there is no simple alternative assumption we could make that would provide a sensible alternative; for instance gamma or Poisson distributional assumptions would not work here as the distribution is not skewed in a manner that would render these methods better than normality. Assumption ranking alone of scores and basing analysis on ordinality alone is possible (e.g. Parsons NR. Proportional-odds models for repeated composite and long ordinal outcome scales. *Statistics in Medicine* 2013; 32:3181-3191) but that is probably beyond the scope of this current manuscript.

VERSION 2 – REVIEW

REVIEWER	Matthias Knobe RWTH University Aachen Dpt. of Orthopaedic Trauma Germany
REVIEW RETURNED	10-Dec-2017
GENERAL COMMENTS	Thank you for your effort to revise the manuscript. All of my issues are addressed now and I recommend the publication. Congratulation for this good work.